# Flexural Performance and Microstructural Studies of Trough-Shaped Geopolymer Ferrocement Panels

**DOI:** 10.3390/ma15165477

**Published:** 2022-08-09

**Authors:** Malathy Ramalingam, Poornima Mohan, Parthiban Kathirvel, Gunasekaran Murali

**Affiliations:** 1Department of Civil Engineering, Sona College of Technology, Salem 636005, Tamil Nadu, India; 2School of Civil Engineering, SASTRA Deemed University, Thanjavur 613401, Tamil Nadu, India; 3Peter the Great St. Petersburg Polytechnic University, 195251 St. Petersburg, Russia

**Keywords:** geopolymer mortar, ferrocement panel, microstructural analysis, flexural behavior

## Abstract

Geopolymer mortar is the best solution as an alternative to cement mortar in civil engineering. This paper deals with the effect of geopolymer mortar on the strength and microstructural properties under ambient curing conditions. In this research, geopolymer mortars were prepared with fly ash and steel slag (in the ratio 1:2.0, 1:2.5 and 1:3.0) as precursors with NaOH and Na_2_SiO_3_ as activator solution solutions (in the ratios of 0.5, 0.75 and 1.0) with concentrations of NaOH as 8 M, 10 M, 12 M and 14 M to study the compressive strength behaviour. From the experimental results, it was observed that the geopolymer mortar mix with the ratio of fly ash and steel slag 1:2.5, 12 M NaOH solution and the ratio of NaOH and Na_2_SiO_3_ 0.5 exhibits the maximum compressive strength results in the range of 55 MPa to 60 MPa. From the optimized results, ferrocement panels of size 1000 mm × 1000 mm × 50 mm were developed to study the flexural behaviour. The experimental results of the flexural strength were compared with the analytical results developed through ABAQUS software. It was observed that the Trough-shaped geopolymer ferrocement panel exhibits 56% higher value in its ultimate strength than the analytical work. In addition to the strength properties, microstructural analysis was carried out in the form of SEM, EDAX and XRD from the tested samples.

## 1. Introduction

The development of geopolymer mortar, instead of conventional cement mortar, is found to be strengthening the environmental health. The properties of geopolymer mortar give desirable results in comparison with the conventional cement mortar when prepared under suitable conditions [1]. In the construction sector, researchers have to find the essential factors to use clean energy in affordable ways for finding alternatives to the cement industries by way of taking into account of several environmental factors [2].

The varieties of relieving temperature are primarily affected by the physical and compound characteristics of fly ash just as the structure of substance of the activators are. The temperature and term of warmth restoring assume a significant factor for the quality improvement of fly ash-based geopolymer mortar. Geopolymer concrete can also be used for self-compacting concrete, and it doesn’t require vibration for placing concrete when using low-calcium fly ash-based self-compacting. As water exceeds 12% by mass of fly ash in self-compacting geopolymer concrete, it decreases the compressive strength [3]. 

Geopolymers are a type of binder which can be developed with a source material that are rich in silicon (Si) and aluminum (Al), such as fly ash, slag, etc. that can be activated with alkaline solution, primarily the combination of sodium hydroxide (NaOH) and sodium silicate (Na_2_SiO_3_) solution. Unlike conventional Portland/pozzolanic cements, geopolymers don’t constitute calcium-silicate-hydrates (C-S-H) for network arrangement and quality. However, the use of the polycondensation of silica and alumina antecedents was observed in achieving the structural quality. Silica and alumina based material plays an important role in geopolymers, which were broken up in the alkaline based medium, thus steady polymeric systems of alumina silicates are being processed. It was prudent to realize that the calibre and nature of the pre-owned source material would confirm the impact of physical and substance properties of geopolymer concrete. In general, Kaolinite is the regularly utilized crude material because of its high substance of alumina content. Henceforth, it thermally actuated to change into nebulous metakaolinite which was progressively solvent in the fundamental arrangement. Varying the curing temperature such as 80 °C, 100 °C and 120 °C for the period of 4, 6 and 20 h in the process of making a ratio of 1:2 alkaline solution of geopolymer mortar gives higher compressive strength @ 120 °C within 20 h of curing [4]. 

The mechanical properties of geopolymer mixes depend on various parameters such as the type of precursor used, the amount of precursor employed, the concentration of NaOH solution, the ratio of Na_2_SiO_3_ to NaOH solution, the liquid-binder ratio, the type of curing regime, etc. Most of the previous literature utilized low calcium fly ash as a source material in the production of geopolymer binders, while few works have addressed the utilization of slag as a partial replacement for fly ash [5] and other source materials. The use of blast furnace slag in geopolymers enhances the mechanical properties, and the mixture of 60% metakaolin and 40% blast furnace slag shows a denser structure with improved mechanical properties [6]. Hence, this study has been focused on the replacement of fly ash with slag at the ratio of 1:2.0, 1:2.5 and 1:3.0. With respect to the NaOH concentration, the mechanical properties of geopolymer mixes increase with the increase in the NaOH concentration [7], and most of the previous literature has taken the NaOH concentration to a maximum of 16 M and observed that 12 M to 14 M NaOH solution mix results in optimum mechanical properties. A higher concentration of geopolymer concrete activated by 12 M NaOH solution gives higher strength than the mix with 8 M NaOH concentration [8]. Hence, this research has been carried out with NaOH concentrations of 8 M, 10 M, and 12 M, with a maximum concentration of 14 M. In terms of the NaOH to Na_2_SiO_3_ ratio, the past studies have suggested the maximum volume of Na_2_SiO_3_ solution than NaOH solution. An increase in the dosage of alkaline solution increases the pH value of the slag-based geopolymer mortar mix and with an addition of 5% and 8% of Na_2_O leads to less deterioration under a chloride environment [9]. Hence, this study has been taken in to account of the ratio of NaOH to Na_2_SiO_3_ as 0.50, 0.75 and 1.00. In terms of the liquid-binder ratio, increasing the water/binder ratio increases the ease to work but affects the strength of the specimen [10].

In terms of curing regime, plenty of researchers have focused on elevated temperature curing rather than conventional water curing or ambient temperature curing. Geopolymer mixes produced with 12 M NaOH solution, Na_2_SiO_3_ to NaOH ratio of 2.5 and liquid-binder ratio of 0.45 cured at 90 °C for 12 shows optimum compressive strength results and was found to decrease at 80 °C and 100 °C after curing for 24 h [11]. It was observed that the 70% of 28 days compressive strength of the geopolymer mixes can be achieved within three to four hours of elevated temperature curing at 65 °C, and it was also concluded that there were no significant variation in the compressive strength after 28 days of curing [12]. Hence, elevated temperature curing may be employed in circumstances where instantaneous compressive strength is required. In addition, elevated temperature curing may be adopted for precast construction and may not be suitable for conventional cast in situ conditions. Also, in tropical countries such as India, the temperature may go up to 40 °C and, hence, elevated temperature curing may not be a compulsive factor. Therefore, this study has been carried out to evaluate the effectiveness of the ambient temperature curing condition so as to apply it in cast in situ conditions.

The variation in compressive strength of the geopolymer mortars depends on the particle size distribution, shape and texture. The density and compressive strength of the geopolymer mortar prepared with natural sand has improved performance over the mix prepared with quarry dust and manufactured sand. In addition, the mixes prepared with partial replacement of natural sand quarry dust achieves a comparable performance [13].

When exposed to 5% solutions of HCl and H_2_SO_4_ and their combinations, fly ash based geopolymer mortars show less deterioration than the conventional cement mortar, which signifies its superior performance under an acid medium [14] and chemical attacks [15]. After the sintering process, the size and number of pores in kaolin-based geopolymer significantly affects the density of material and water absorption which indicates the presence of pores in the material, which was also in line with the tomography imaging [16]. The use of recycled concrete aggregate as a partial replacement for natural coarse aggregate has a significant effect on the engineering, durability [17] and flexural behaviour [18] of slag-based geopolymer concrete mixes. Use of 1% steel fibre and 0.5% sisal fibre helps in increasing the tensile strength characteristics of geopolymer mortar [19].

Geopolymer mortars can also be effectively used in the process of repairing works due to its amazing mechanical properties. A mix with a binder to sand ratio of 0.5 has been adopted for concrete patch repair [20]. Flyash and GGBS-based geopolymers have also been used for soil stabilization, particularly for the applications of road construction in Subgrade layers [21].

Ferrocement has the thin wall reinforced concrete element which has the composite material that comprising thin reinforcement with cement mortar. It will be light in weight and has good resistance against cracking. It can be folded to any shape as it does not require skilled labor, expensive material or more time for construction [22]. The ductility of the panels becomes enhanced when extending the mesh layer, particularly at the time of using chicken mesh in the ferrocement panels [23]. In recent times, alkali-activated binders (AABs) were the most used essential material for the replacement of portland cement. To transform the portland cement manufacturing industry into an AAB factory did not require a huge investment for the large scale production of AABs. Therefore, in the current circumstances it is required to explore the standardization formulas for the commercial development of AABs in the construction sector [24]. It has been observed from the Feret equation that a significant amount of cement is needed to achieve the same compressive strength as that of the alkali-activated concrete. One-part geopolymer concrete produces less than 10% of global warming potential (GWP) than concrete made with 100% ordinary Portland cement (OPC), and therefore results in lesser carbon footprint values than the concrete made with OPC [25]. In addition, the strength of the geopolymer mixes has a direct influence on the quantum of calcium in the precursor used [26], and recent works have focused on the utilization of ground granulated blast furnace slag as a precursor. However, the availability of low calcium fly ash is predominant in India, thus this work has been framed with low calcium fly ash as a precursor in the production of geopolymer mixes.

Most of the previous studies have utilized low calcium fly ash as the source material in the production of geopolymer binders, with few works addressing the utilization of slag as a partial replacement for fly ash [5] and other source materials. Hence, this study has been focused on the replacement of fly ash with slag at the ratio of 1:2.0, 1:2.5 and 1:3.0. With plenty of variation with respect to the activator solution concentration and ratio, this work intends to check the effect of NaOH concentrations (8 M, 10 M, 12 M and 14 M) and the ratio of NaOH to Na_2_SiO_3_ as 0.50, 0.75 and 1.00 to evaluate the effectiveness of the ambient temperature curing condition so as to apply it in cast in situ conditions. In addition, this study will focus on making innovative versatile trough-shaped geopolymer ferrocement panels under the flexural load by experimental and analytical methods. Geopolymer ferrocement panels are eco-friendly and sustainable, providing many benefits in the construction industry. The primary objective is to investigate the flexural behavior of geopolymer ferrocement panels using industrial waste such as fly ash and steel slag. 

## 2. Materials and Methods

### 2.1. Materials

#### 2.1.1. Fly Ash

Low calcium fly ash which are rich in Alumina(Al) and Silica(Si) was procured from the Mettur thermal power plant located in Salem, Tamilnadu, India to prepare the geopolymer mortar mix. As per ASTM, the physical properties of fly ash were studied and the results are as follows: specific gravity—3.64, fineness modulus—3.924 and bulk density—1446.93 kg/m^3^. The chemical composition of fly ash was obtained from the manufacturer and the values are as follows: loss of ignition (L.O.I)—1.96%, silica (SiO_2_)—72.10%, calcium oxide (CaO)—1.61%, magnesium oxide (MgO)—1.62%, iron oxide (Fe_2_O_3_)—1.14%, and aluminum oxide (Al_2_O_3_)—19.84%. In addition to the physical properties and chemical composition, the microstructural characteristics of the fly ash sample was investigated with the help of SEM, EDAX and XRD analysis.

#### 2.1.2. Steel Slag

Steel slag was obtained from the steel manufacturing industry, JSW Salem, Tamilnadu, India. The physical properties of steel slag were studied in the laboratory with the standard procedure recommended by ASTM, and the obtained results are: specific gravity—3.72, fineness modulus—4.098, and bulk density—1448.09 kg/m^3^. The chemical composition of steel slag were obtained from the manufacturer and its values are: CaO—46.17%; MgO—10.02%; SiO_2_—17.92%; Al_2_O_3_—6.73%; MnO—1.89%; Fe_2_O_3_—16.98%; P_2_O_5_—1.51%; Na_2_O—0.23%; K_2_O—0.10%; and SO_3_—0.67%. Similar to fly ash, steel slag was also investigated with the help of SEM, EDAX and XRD analysis to study its microstructural characteristics.

#### 2.1.3. Alkaline Solution

Alkaline solution plays a significant role in the geopolymerization reaction process during polymerization. The alkaline liquid was soluble in alkali metals, usually sodium or potassium-based material. The alkaline liquid used in this investigation is a combination of sodium silicate and sodium hydroxide solution. The sodium hydroxide used in this study was 98% pure and its specific gravity was 2.13. The chemical composition of sodium silicate solution was Na_2_O—23.3%; SiO_2_—20.8%; and water—55.9% by mass. The alkaline solution was prepared with sodium hydroxide and sodium silicate at different ratios (0.50, 0.75 and 1.00), with the concentration of sodium hydroxide at 8 M, 10 M, 14 M and 16 M. The sodium hydroxide solution was prepared 24 h before casting to reduce the effect of high temperature evolution.

#### 2.1.4. Chicken Mesh

Chicken mesh was used in this research and was obtained from the market in the form of a roll of hexagonal opening (as shown in Figure 1) of width 1 m, mesh opening 15 mm × 10 mm with thickness at joint as 1.5 mm, diameter of wire as 0.5 mm, unit weight of 1.85 kg/m^2^, density of 7850 kg/m^3^ with an yield strength of 312 MPa. The mesh was cut into the required folded shape and placed in the 8 mm skeleton steel to obtain the correct form. Four layers of chicken mesh were tied to the reinforcement so as to be used in the panel.

### 2.2. Methods

#### 2.2.1. Compressive Strength of Geopolymer Mortar

As there were no standard codal provisions for arriving the geopolymer mixes, various trail mixes were prepared and evaluated to arrive at the optimal geopolymer mortar mix. Cubes of 70.7 mm × 70.7 mm × 70.7 mm size as per IS 516:1959 [27] were used for the estimation of compressive strength results at the age of 3, 7, 14 and 28 days cured under an ambient temperature condition. The specimens were tested under a 200 T capacity compression testing machine with digital reading and stacking speed control with a pace of 0.05 MPa/s while testing.

#### 2.2.2. Microstructure Studies of Geopolymer Mortar

SEM was the most widely used surface analytical technique to observe the morphological characteristics of various materials present in the samples. The optimized geopolymer mortar after 28 days of curing has been taken to investigate the morphological characteristics using XL30 SEM instrument. The range of scale used in the SEM analysis was 5 µm with the resolution of 5000×. EDAX is an analytical technique that can focus on characterization in the chemical zone in addition to the analysis done at the element level. It relies on the interaction of X-ray source excitation and materials. The elemental composition of materials was identified for the 28 days-cured optimized geopolymer mortar sample. The XRD was plotted on a graph between 2Ɵ and intensity, where 2Ɵ have been in the *x*-axis and the intensity in the *y*-axis, where 2Ɵ is the angle between the transmitted beam and reflected beam of the samples. An X-ray diffractometer with a D8 advance model made by Bruker (Karlsruhe, Germany) which has a tube voltage of 2.2 kW Cu-anode ceramic tube and a scan step length of 0.02° was used.

#### 2.2.3. Details of the Geopolymer Ferrocement Panels

The geometry of the geopolymer ferrocement panels was 1000 mm × 1000 mm × 50 mm, which was adopted with the help of standard design principles and the detailed sketch of the panel is given in Figure 2 (as per ACI 549.IR-93 [28], reapproved 1999). Four layers of the chicken mesh were embedded with the geopolymer ferrocement panels with the help of 8 mm skeletal steel to withstand the ductile nature of the specimen. The cross-section of the panels is shown in Figure 3.

Figure 4 shows the three dimensional view of the geopolymer ferrocement panel with the first layer provided with optimized geopolymer mortar over a thickness of 25 mm and the 8 mm skeletal reinforcement placed over the mortar. This skeletal steel was embedded in such a manner that it satisfies the required shape mounted with the specified layers of the chicken mesh with four in numbers in the present case, and the final layer holding the Geopolymer mortar with a thickness of 25 mm has been placed.

#### 2.2.4. Preparation of Trough-Shaped Geopolymer Ferrocement Panels

Initially, the moulds have been prepared in the optimum trough-shaped structure with the help of a steel sheet. During casting, the prepared optimized geopolymer mortar with a thickness of 15 mm was spread evenly over the base, and four layers of prepared chicken mesh combined with the skeleton rod were then placed over the mortar. This was again covered with the geopolymer mortar over a total thickness of 50 mm. Mesh preparation details, the casting of the geopolymer ferrocement panel and their test setup are shown in Figure 5a–d. 

The specimens were tested under 200 T Loading frame machinery with lateral support and a hydraulic jack of 2000 kN capacity with a gauge and hand pump equipped with a demountable mechanical strain gauge (100 mm) of least count 0.01 mm. The edges of the panels were supported on the roller and the schematic view of the test setup is shown in Figure 5d. The strain gauges were placed at the midspan of the panel to note the strain readings. The load and the corresponding deflections were measured continuously until the panels reached the ultimate load.

#### 2.2.5. Preparation of Multipurpose Geopolymer Ferrocement Panels

Trough-shaped geopolymer ferrocement panels were prepared and assembled in order to achieve multiple purposes. One panel was cast and finished in the middle, leaving the sides for welding with the other two panels, and the remaining two panels were cast in 3/4th so that these three were merged, as shown in Figure 6a,b. As per IS 456 [29], lap distance for the slab have been provided as 30d, where ‘d’ is the diameter of the skeletal steel (8 mm in the present study). A lap distance of 240 mm (30 × 8 = 240 mm) was provided for combining the panels. 

#### 2.2.6. Flexural Behavior of Geopolymer Ferrocement Panel

In this research work, the flexural behavior of a geopolymer ferrocement panel has been comprehensively investigated. The experiments include the load-deflection behavior, maximum load carrying capacity and the crack pattern of the panels. The load was transferred through two point loading conditions and the least count of the strain gauge used in the investigation was 0.01 mm.

#### 2.2.7. Analytical Work of Geopolymer Ferrocement Panel

The analytical study of the geopolymer ferrocement panels has been carried out using ABAQUS Software. Each section was modelled with the aid of ABAQUS software, which can then be extruded in any direction and the 3D solid element in modeling space using deformable type for panels was been created. Finite element models with three dimensions of ferrocement panels were developed, and the various items concerned with modeling were addressed as follows: element type, specify boundary conditions, material assigning property, loading conditions, meshing operations, assigning sections and finally results analysis were performed to complete the entire analytical process. The element thus utilized was capable of undergoing plastic deformation with a crack response in three orthogonal directions and it therefore enables the crushing response.

## 3. Results and Discussion

### 3.1. Compressive Response of Geopolymer Mortar

Figure 7, Figure 8 and Figure 9 show the variation in the compressive strength results of the geopolymer mortar at 3, 7, 14 and 28 days of ambient temperature curing with variation in NaOH concentrations (8 M, 10 M, 12 M and 14 M), ratio of NaOH and Na_2_SiO_3_ (0.50, 0.75 and 1.00) and the ratio of fly ash and steel slag (0.50, 0.75 and 1.00), respectively. It was observed from the experimental results that the compressive strength of the geopolymer mortar increases with the increase in the NaOH concentration irrespective of curing age. The sodium hydroxide concentration in the aqueous environment of the geopolymeric system affects both the dissolving mechanism and the bonding of solid particles in the concluding structure. Since silica and alumina are more thoroughly leached when using a high concentration NaOH solution, the solid components dissolve more quickly and the geopolymerization process results in increased compressive strength [30]. Even with a 14 M NaOH concentration, the mix results in the highest compressive strength results, the work was further taken with 12 M NaOH concentration mix, as this mix results in the required compressive strength results at the age of 28 days curing.

With respect to the ratio between NaOH and Na_2_SiO_3_, as shown in Figure 8, the compressive strength results were found to decrease with the increase in the ratio of NaOH and Na_2_SiO_3_. This might be due to the availability of additional silicate ions which results in the development of an additional geopolymerization reaction which results in improved compressive strength results with the reduction in the ratio of NaOH and Na_2_SiO_3_. Regarding the replacement level of fly ash with steel slag, the compressive strength results were found to be optimum at the ratio of 2.5 as opposed to 2 and 3, irrespective of the age of curing; which is evident from Figure 9. Among all of these results, a flyash and steelslag ratio of 1:2.5, a ratio of NaO and Na_2_SiO_3_ of 0.5 and NaOH concentration of 12 M gives 56 MPa compared to 50 MPa and 40 MPa for a 1:2 and 1:3 ratio, respectively. The compressive strength attained at the 14th day was about 98% of its 28 day compressive strength results, which signifies that the target strength has been achieved before 28 days of curing. The compressive strength of the geopolymer mortar depends on the angularity of raw materials, surface texture, surface area and the bond between alkaline solutions and raw materials. The angular particles of steel slag, its high surface area and the bond between alkaline solutions resulted in higher compressive strength.

### 3.2. Microstructural Analysis

#### 3.2.1. Scanning Electron Microscope Analysis

The SEM morphology of fly ash shown in Figure 10 indicates that it was spherical in shape and more compact than the steel slag, which is evident from the SEM image shown in Figure 11. Slag particles are dense angular, irregular, and porous in shape, and it helps to strengthen the bond between fly ash and steel slag. Many researchers have proved that these angular shapes enhance the bonding properties [9,31,32,33,34]. Figure 12a,b displays the SEM image of optimized geopolymer mortar, which has a homogeneous structure with an embedded matrix formation. The dense structure manifests the geopolymer mortar exhibiting stronger cementing between fly ash and steel slag. It leads to a better result, and a similar observation was made by Abideng HAWA [35]. The original morphology changes due to the hydration process that tends to occur in geopolymer because of the formation of CSH gel.

#### 3.2.2. EDAX (Energy Dispersive X-ray Spectroscopy)

Figure 13 shows the EDAX pattern of the fly ash which contains crystalline peaks for the mineral oxides, alumina and silica. Similarly, Figure 14 shows the EDAX pattern of the steel slag, which shows evidence of the traces of silicon, aluminum, iron, magnesium and calcium. Figure 15 shows the EDAX pattern of the optimized geopolymer mortar after 28 days of curing, which indicates crystalline phase materials with obvious quantities of alumina, silica, magnesium, oxides and ferrous.

#### 3.2.3. XRD (X-ray Diffraction)

The XRD patterns of the fly ash sample shown in Figure 16 contains a series of crystalline phases such as quartz (SiO_2_), mullite (Al_2_O_3_), hematite, and kaolinite (3Al_2_O_3_·2SiO_2_), thus mainly shows the specific compounds of alumina and silica. Figure 17 shows the XRD patterns of the raw steel slag powder sample, which shows the evidence of quartz and kaolin formation in the crystalline form. Similarly, the X-ray diffractograms of the optimized geopolymer mortar mix after 28 days of curing is shown in Figure 18. It has been observed from the XRD patterns that the peaks were found to be in the range of 25° to 68° 2Ɵ region as a result of geopolymer characterization. 

The XRD patterns show the evidence of the formation of crystalline phases of quartz, calcium silicate hydrate (C-S-H), sodium alumino silicate gel, jaffeite (Ca_4_(Si_3_O_7_)(OH)_6_) and calcite (CaCO_3_), as shown in Figure 18 [16,36]. The overlapping mineral phases in the steelslag could not be determined with certainty. The slight humps observed in the region of 20 to 60 2Ɵ indicated the presence of amorphous nature as a result of the geopolymerization process. The peaks of XRD of flyash are different from the XRD of optimized geopolymer, which indicated the formation of amorphous structures in the geopolymer. The crystalline peaks C-S-H, sodium alumino silicate gel and Jaffeite are evidence for the improved strength properties of geopolymer mortar and dense microstructure, as evident from the SEM micrographs, which help in the formation of geopolymerization and leads to an increase in strength; this was observed in other studies [37,38] as well. From Figure 18, the formation of C-S-H type gels was observed due to the additional hydration process, which helps in improving the compressive strength, which was also presented by [36,39]. The formation of calcite tends to increase the alkaline nature of the geopolymer mortar, thereby improving strength properties.

### 3.3. Flexural Behaviour of Geopolymer Ferrocement Panels

#### 3.3.1. Load-Deflection Behaviour

For the geopolymer ferrocement panels, the load-deflection behavior has been considered to be an essential feature. The panels cast were tested under the two-point loading conditions; their load and the corresponding deflection values for the average of three panels were recorded and depicted in Figure 19. As the load increases gradually, the deformation increases, which results in the widening of the crack widths, and as the load increased further, many cracks were observed. The ultimate load of the trough-shaped multipurpose geopolymer ferrocement was recorded. The load-deflection characteristics of the developed model using ABAQUS software is also projected in Figure 19 as a comparison for the experimental results. From the graph, it can be seen that there is a slight difference between the load-defection curve obtained from experimental and analytical work, where the deflection value was measured to be 10 mm in analytical work, which is higher than the experimental work, where the deflection was measured to be 8 mm. With the increase in the load, the analytical curve shows a higher load (52 kN) carrying capacity than the curve obtained from the experimental results, until it reaches failure. The deflection of 8 mm and the corresponding load carrying capacity of 41 kN was obtained in the experimental analysis.

#### 3.3.2. Crack Pattern

Crack patterns have been carefully observed and analysed throughout the loading process. Under a flexural load, the geopolymer ferrocement panels have undergone three stages: the initial cracking stage, the post cracking stage, and the post-yielding stage. Well-distributed cracks have been recorded from the experimental results. At the initial stage, fine cracks were found to appear at the bottom of the panel, and after increasing the load, the vertical crack pattern increases and extends from the bottom to the top of the specimen. These cracks were noted until they reached the ultimate load. A schematic view of the crack pattern is shown in Figure 20. To study the flexural behaviour of panels, it is necessary to study the load-deflection curve, first crack, ultimate load and maximum deflection characteristics. Therefore, the comparison of both analytical and experimental work was considered to be very effective one. Initially, cracks were observed in the tension zone, which is shown in Figure 20, which is also evident from the analytical results as shown in Figure 21. The chicken mesh reinforcement yields were first followed by the crushing of mortar in the specimen. 

### 3.4. Analytical Study of Geopolymer Ferrocement Panels

In order to develop geopolymer ferrocement panels, 12-node continuum solid element was utilized. The solid element has 12 nodes with three degrees of freedom at each node: translations in the nodal x, y, and z directions. In consideration of material properties, mortar, mesh and steel properties have been dealt in terms of density, elastic and plastic behaviours. The mortar part was designed by 3D deformable solid extrusion type. 

In addition to the mesh layers, an 8 mm skeletal rod has been provided to improve the load carrying capacity of the panel. Then, mortar and mesh have been assembled as shown in Figure 22a,b, respectively. Suitable governing equations were reckoned for the assumed element model and assigned to nodes in terms of mathematical expressions, as meshing leads to accuracy in the finite element approach. The revealing fact could be such that an increase in the element volume will lead to the accuracy in calculation and thus the accuracy of results will be obtained while being compared with other finite element-based software.

Load was given as the point load using the interaction module. The created point load was applied on the top surface of the panel at each node. The load was evenly distributed on the selected node. Hinged supports were given on both the ends of the panel. Figure 22c shows the applied boundary conditions of the ferrocement panel. A job analysis has been done based on the type of formulation considered for the problem identified. The findings were visualized and compared with the experimental results. 

In this research work, the results have confirmed that fly ash-based geopolymer proved to be the best alternative for the cement mortar, and trough-shaped geopolymer ferrocement panels were best suited for sustainable construction. The reaction between silica and alumina was higher with the help of alkaline solutions to tend towards the geopolymerization process which mainly contributes to the improved compressive strength. 

## 4. Conclusions

The compressive strength results of the geopolymer mortar mix increases with the increase in the NaOH concentration, reduces with the increase in the ratio of NaOH to Na_2_SiO_3_, and the optimum replacement level of fly ash with steel slag was observed to be 2.5.The optimum mix proportioning was found to be: fly ash to steel slag ratio of 1:2.5, NaOH/Na_2_SiO_3_ ratio of 0.5 and NaOH concentration of 12M, which results in a 28th day compressive strength of about 56 MPa, which is higher than the corresponding mixes.The test results of SEM, EDAX and XRD show dense homogenous and silicious ingredients in the form of quartz, C-S-H gels and mullite, which enhance the formation of the geopolymerization reaction with improved compressive strength results.About 98% of the 28 days compressive strength values were attained at 14 days of curing, which signifies that the target strength has been achieved before 28 days of curing.Ferrocement panels are crack-resistant due to the delay in the occurrence of their initial crack. Initial cracks occur at a load of 10 kN with a corresponding deflection of 4 mm.It was observed that the trough-shaped ferrocement panels provide improved ultimate strength and crack width of the geopolymer ferrocement panels, and are considerably narrow.The ultimate load carrying capacity of the geopolymer ferrocement panel substantially improved.The experimental results of the trough-shaped geopolymer ferrocement panel exhibits 56% higher load carrying capacity in its ultimate stage than in the analytical work. The crack Pattern obtained in the software correlates effectively with the experimental results.The use of industrial byproducts in the production of geopolymer mortar significantly improves the sustainable nature of geopolymer mixes compared to the conventional cement mortar. The prepared trough-shaped geopolymer ferrocement panel results in improved performance in the analytical results, which shows the effective utilization of geopolymer materials in the production of the ferrocement panel, which results in sustainable construction practices.

## Figures and Tables

**Figure 1 materials-15-05477-f001:**
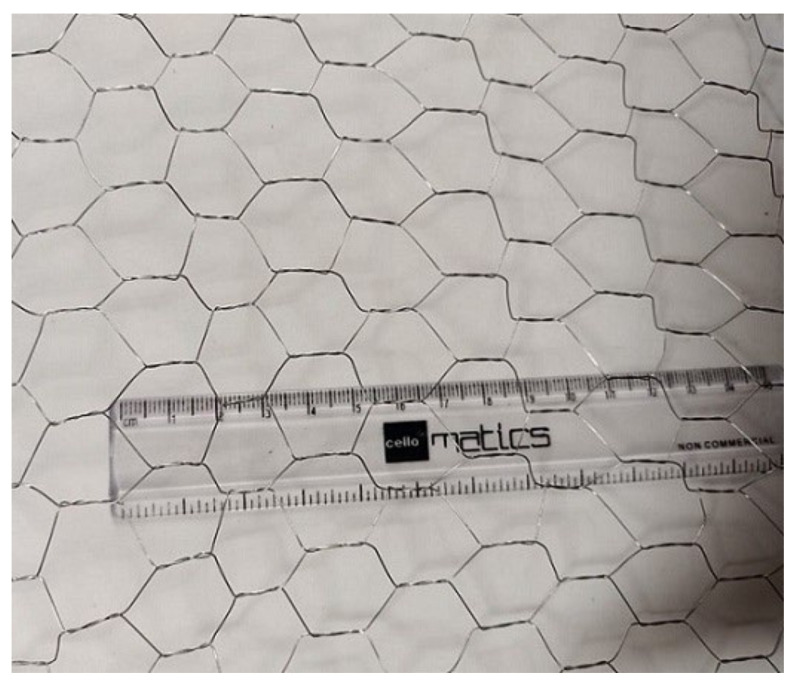
Chicken Mesh.

**Figure 2 materials-15-05477-f002:**
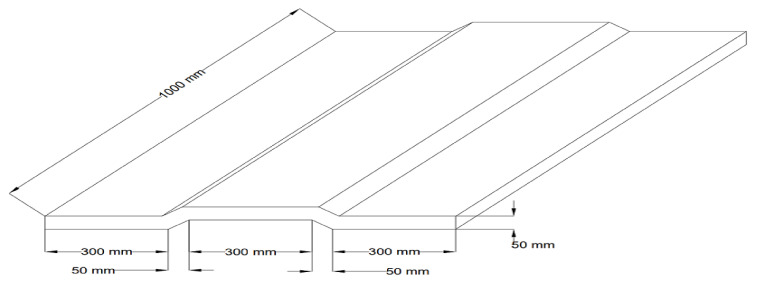
Geometry details of Ferrocement Panel.

**Figure 3 materials-15-05477-f003:**
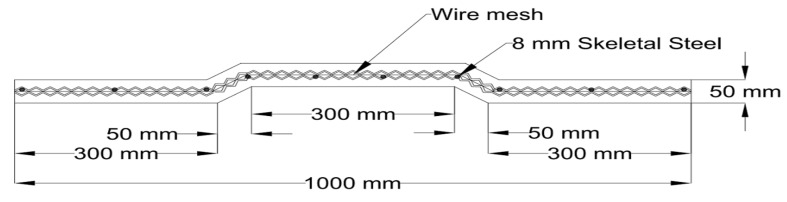
Cross Section view of Panel with four layers of chicken mesh.

**Figure 4 materials-15-05477-f004:**
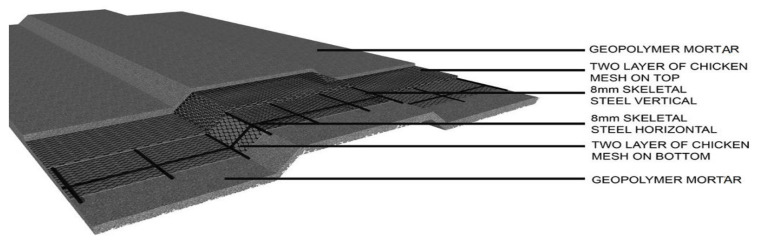
3D view of the Geopolymer ferrocement panel.

**Figure 5 materials-15-05477-f005:**
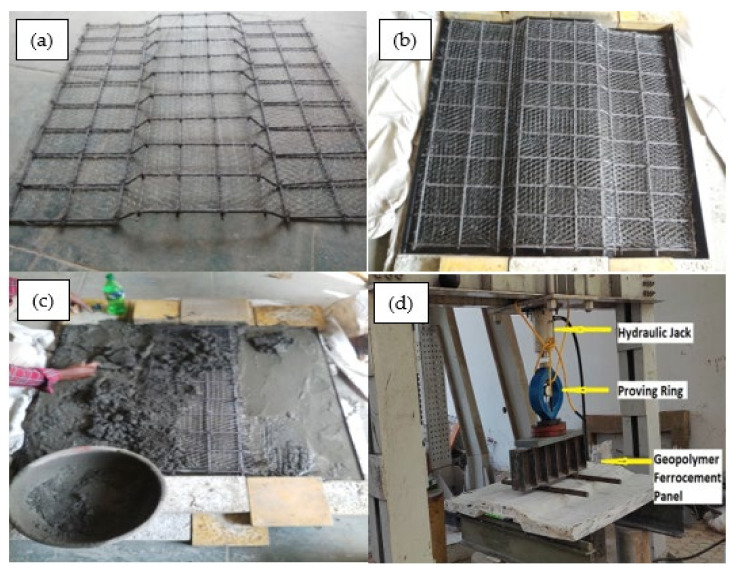
(**a**) Skeletal steel with mesh layers, (**b**) Arrangement on mould, (**c**) Casting of panel, and (**d**) Test setup of panel.

**Figure 6 materials-15-05477-f006:**
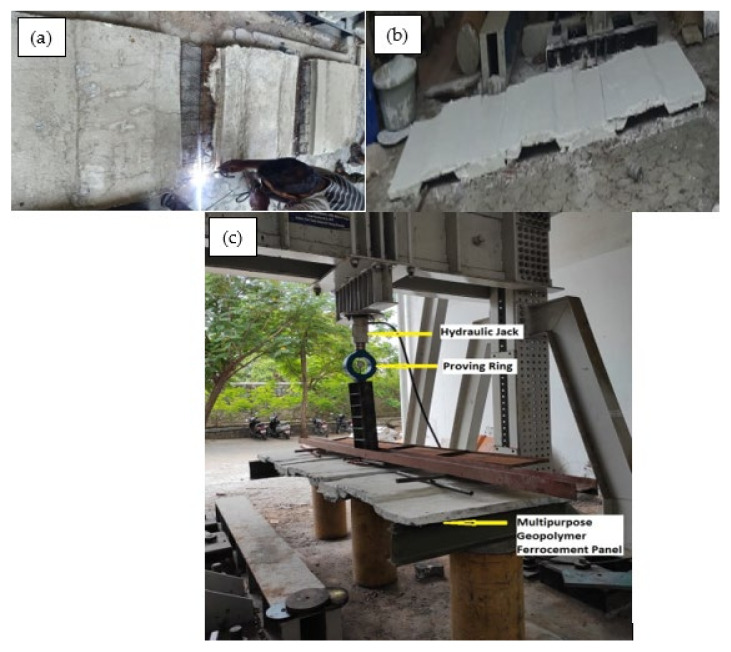
Preparation and testing of multipurpose geopolymer ferrocement panels (**a**) Welding of multipurpose panels, (**b**) Prepared multipurpose panels, and (**c**) Test setup for multipurpose panel.

**Figure 7 materials-15-05477-f007:**
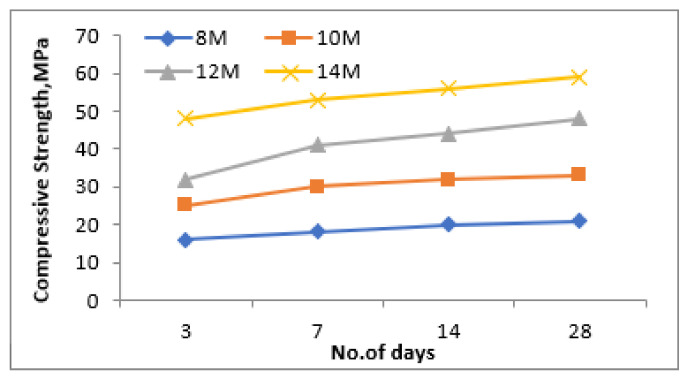
Compressive strength for different molarities of NaOH.

**Figure 8 materials-15-05477-f008:**
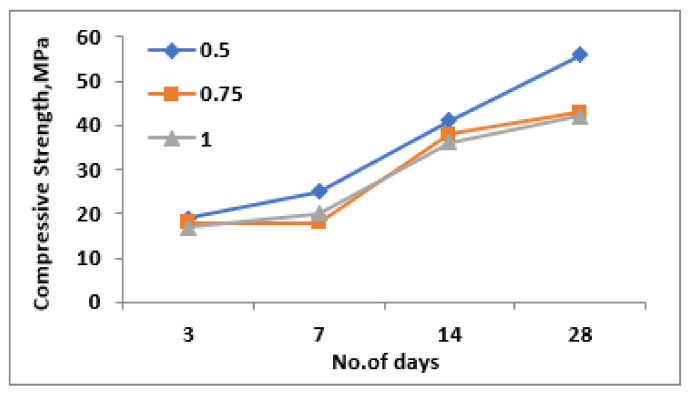
Compressive strength for different NaOH/Na_2_SiO_3_ ratio.

**Figure 9 materials-15-05477-f009:**
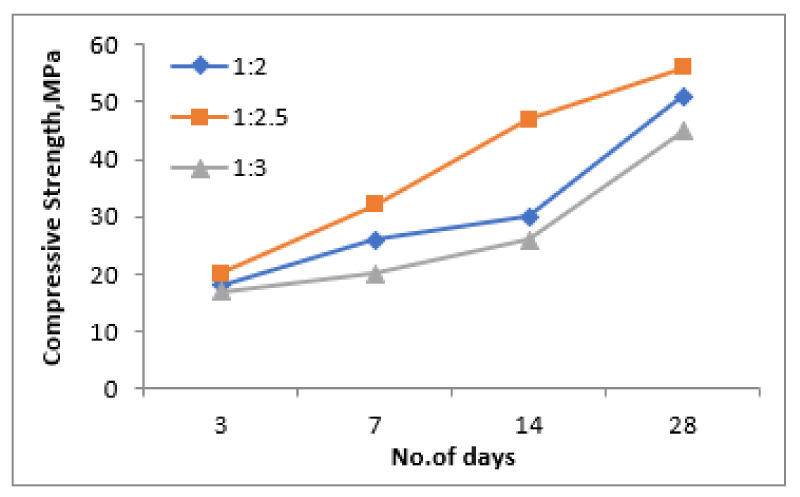
Compressive strength for different fly ash and steel slag ratio.

**Figure 10 materials-15-05477-f010:**
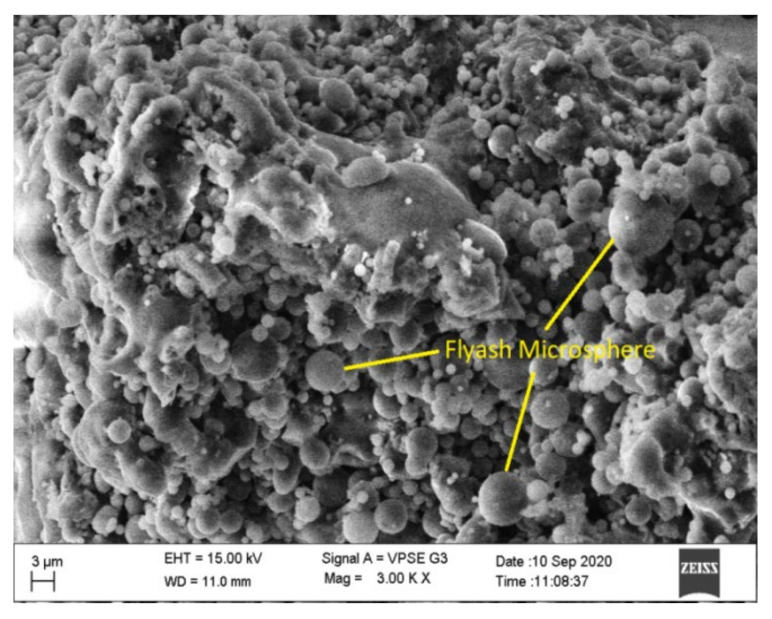
SEM image of fly ash.

**Figure 11 materials-15-05477-f011:**
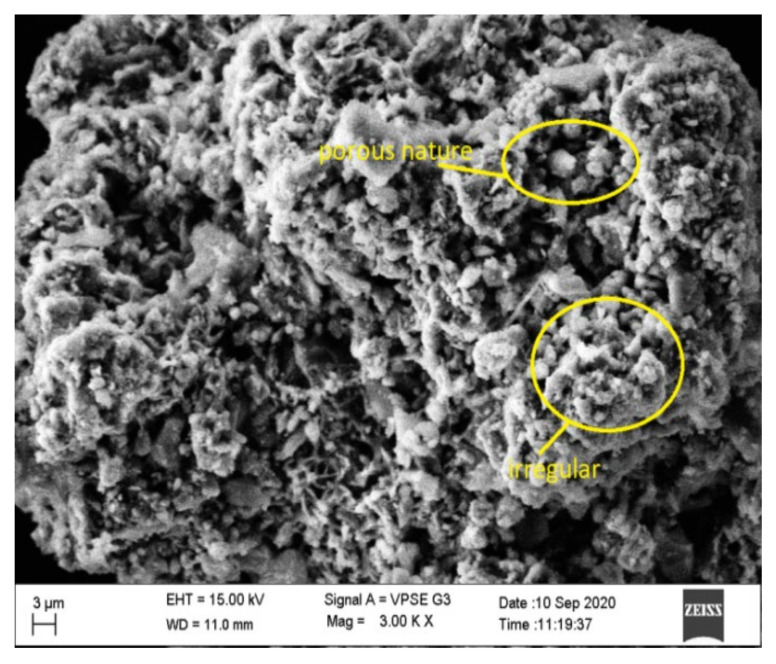
SEM image of steel slag.

**Figure 12 materials-15-05477-f012:**
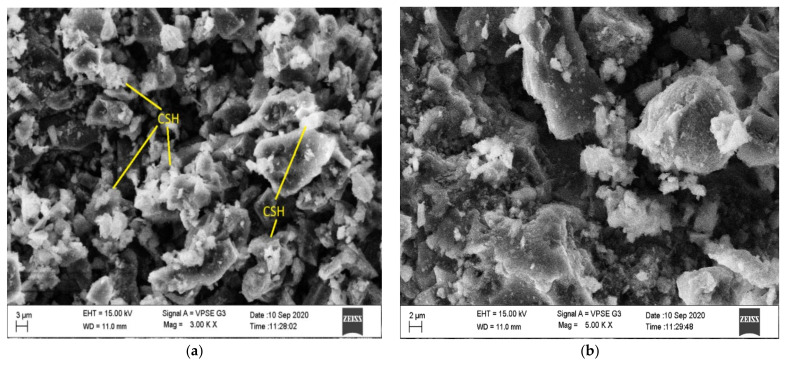
(**a**,**b**) SEM image of optimized geopolymer mortar mix at day 28.

**Figure 13 materials-15-05477-f013:**
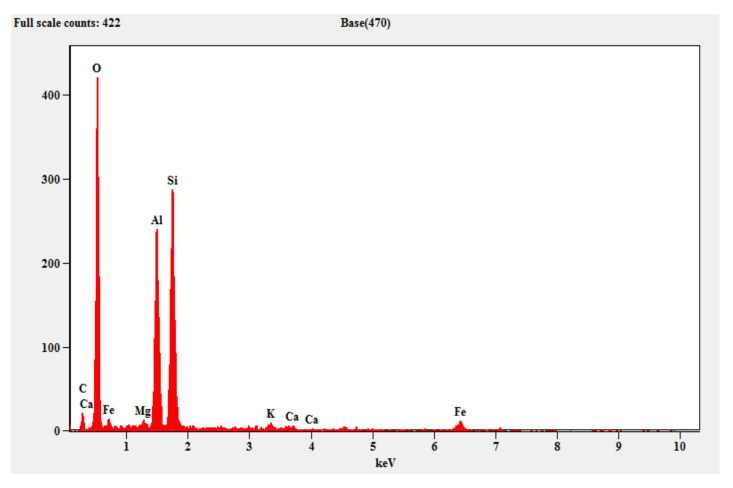
EDAX for fly ash.

**Figure 14 materials-15-05477-f014:**
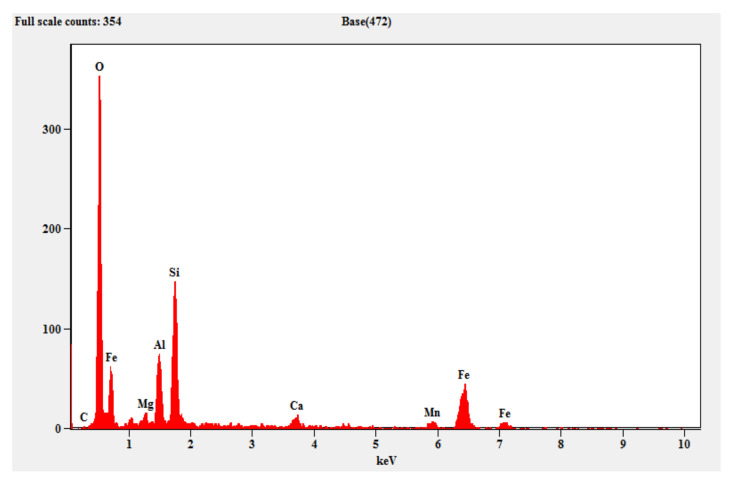
EDAX for steel slag.

**Figure 15 materials-15-05477-f015:**
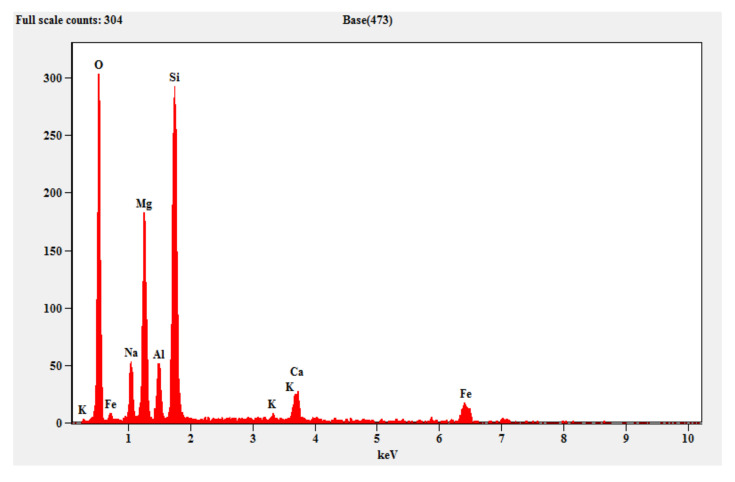
EDAX for optimized geopolymer mortar mix.

**Figure 16 materials-15-05477-f016:**
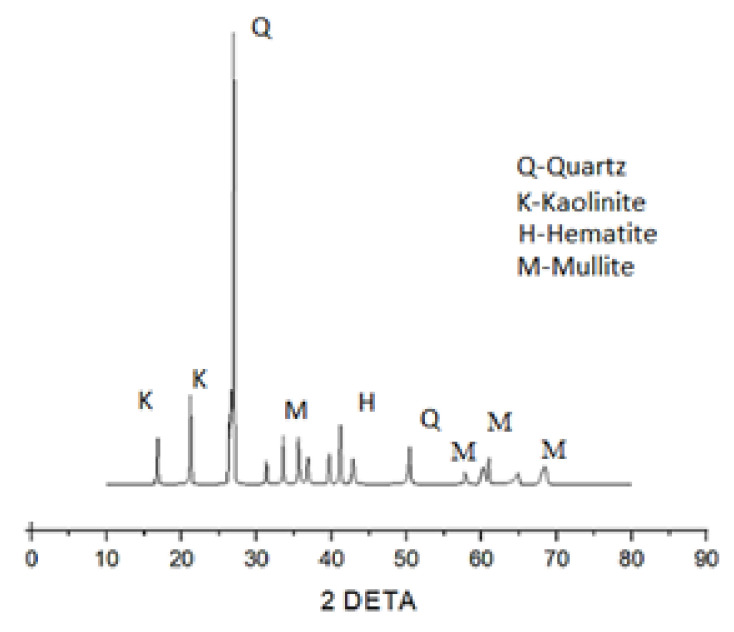
XRD for fly ash.

**Figure 17 materials-15-05477-f017:**
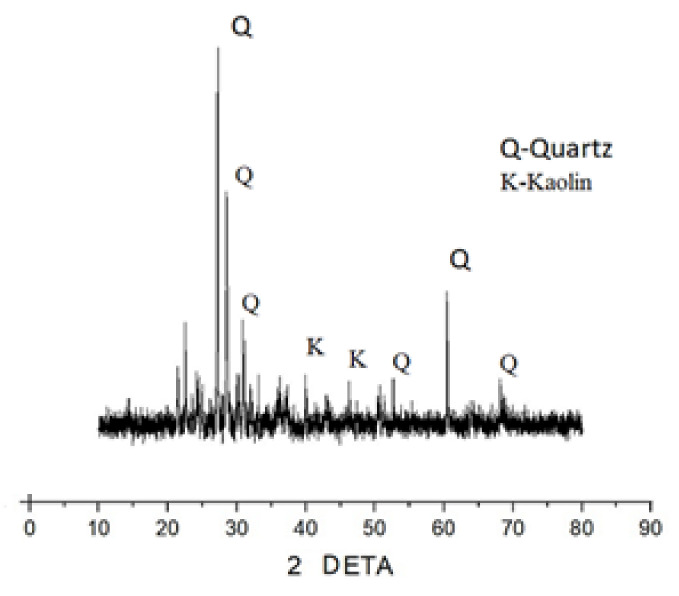
XRD for steel slag.

**Figure 18 materials-15-05477-f018:**
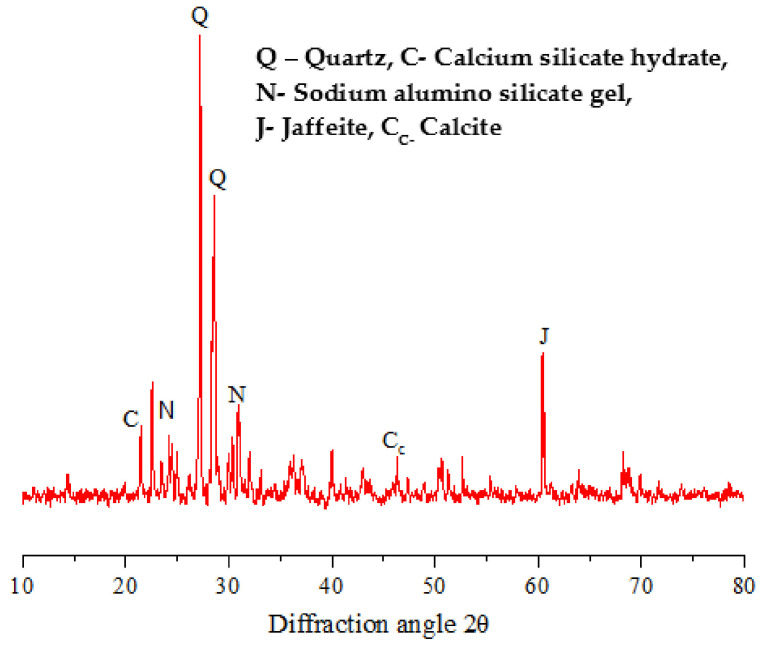
XRD for optimized Geopolymer mortar mix.

**Figure 19 materials-15-05477-f019:**
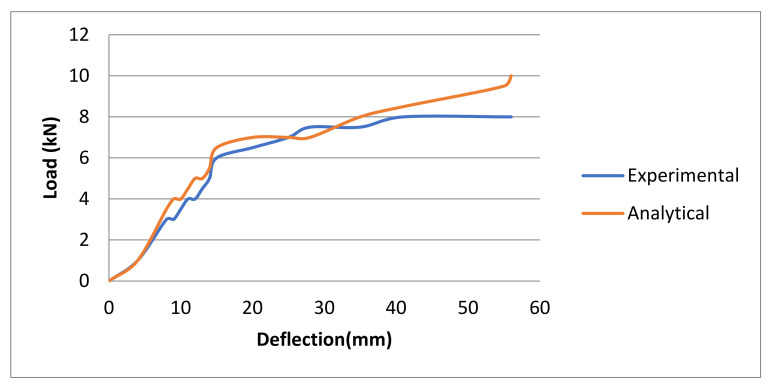
Load vs deflection of optimized geopolymer ferrocement panel using experimental and ABAQUS software.

**Figure 20 materials-15-05477-f020:**
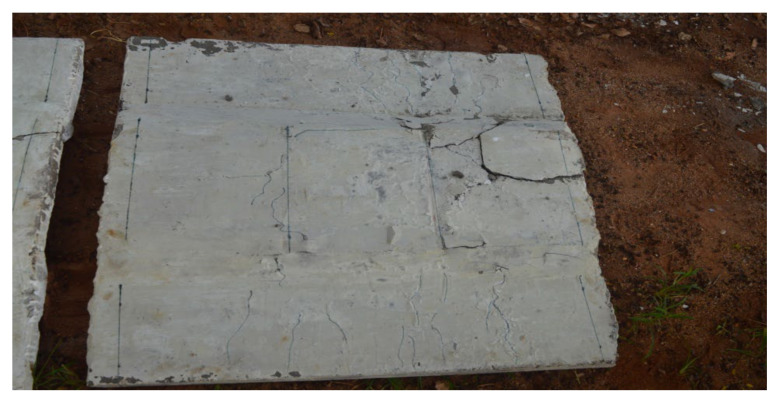
Crack pattern of geopolymer ferrocement panels.

**Figure 21 materials-15-05477-f021:**
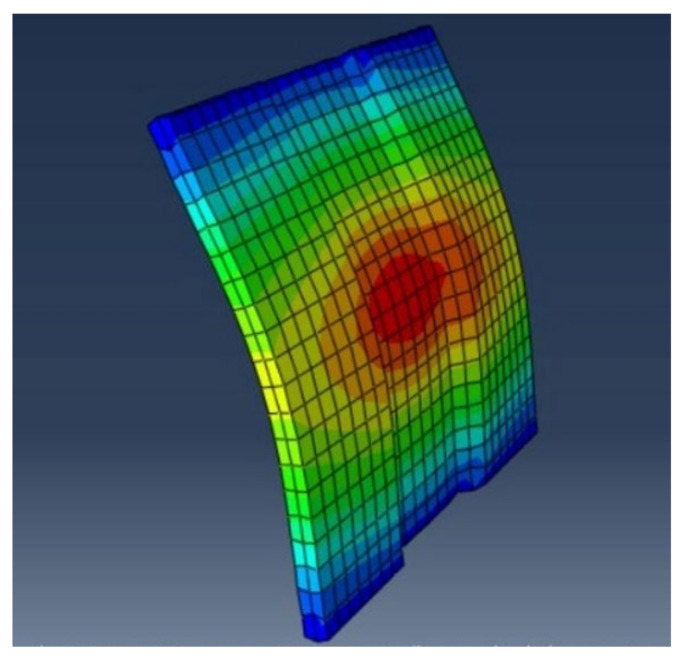
Deformation Pattern using ABAQUS.

**Figure 22 materials-15-05477-f022:**
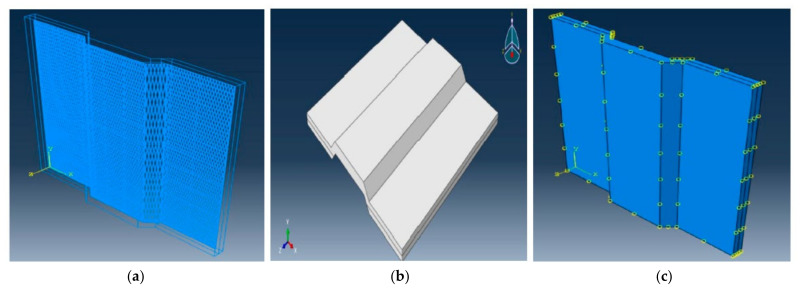
(**a**). Assigning Mesh Detailing, (**b**). Assigning Mortar Property, and (**c**) Assigning Boundary Conditions.

## Data Availability

Not applicable.

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
