# Peer review of "Flexural Performance and Microstructural Studies of Trough-Shaped Geopolymer Ferrocement Panels"

_materials, 2022, doi:10.3390/ma15165477_

Round 1

Reviewer 1 Report

The conducted work “Flexural Performance of trough-shaped Geopolymer Ferrocement panels – Addressing 7,9,11,13 of Sustainable Development Goals” is good. However, following comments should be addressed to further improve paper:

1.      Title should be modified, as there is no discussion in sufficient depth about relation of experimental research with SDGs, particularly after obtaining result outcomes.

2.      Explicitly mention the novelty and research significance of current work in last paragraph of introduction section.  Also, add recent relevant literature review from 2022 papers in introduction section.

3.      Very brief discussion is made in section 3, i.e. results are explained in a descriptive way. In current form, it looks like a lab report. Outcome should be further discussed in detail with scientific reasoning.

4.      Curves presented in figures 19 and 23 should be compared in one graph. Also, figure 23 is not clearly readable.

5.      Optimized material property and panel behavior should be correlated.

6.      A separate brief section (explaining the relevance of this research for practical implementation) may be added before conclusion section.

7.      Conclusions are little long; thus, these should be made brief and to the point. Closing remarks should be added at the end of conclusion section keeping in mind all conclusive bullet points.

8.      English Language should be improved throughout the manuscript.

Author Response

Please find attached the Response to Reviewer Comments

Reviewer 2 Report

The manuscript entitled "Flexural Performance of trough-shaped Geopolymer Ferrocement panels – Addressing 7,9,11,13 of Sustainable Development Goals" presents an experimental study conducted on obtaining and characterization ferrocement panels with geopolymeric concrete. However, the introduction section includes general affirmation without a quantitative evaluation of previous literature, and many other issues must be addressed. The paper needs major revisions before it is processed further, some comments follow:

Abstract: "Fly ash is replaced 100% with cement and steel slag is replaced 100% with fine aggregate" – this statement is ambiguous, please make corresponding corrections. Currently, it can be understood that the paper is referring to conventional concrete.

Could authors better explain the following sentence: "Based on the broad literature survey, various ratios of binder and differ- 17 ent molarities of sodium hydroxide have arrived.”

Lines 18 and 21 are long and unclear – Please present those data in a much more compact form. "In this study NaOH to Na2SiO3 in the ratio of 0.5, 0.75 and 1 with NaOH concentration of 8, 10, 12 and 14 were considered" …..

The abstract is written qualitatively. The majority of the qualitative statements should be modified for quantified result comparisons. Please replace this kind of formulation "for the effective process" and "gives optimum results"  with clear quantitative formulations.

The organization of the abstract is poor. Currently, the following order was presented: Materials, mixtures, results, materials (samples), and methods. Please organize the sentences with clear rationale (materials, mixture, involved techniques/methods, and results).

Introduction Section

The introduction should be significantly improved. There are multiple affirmations without a clear background in the literature. Please provide corresponding citations and references to support the affirmations from this section.

"6.1 percent since 2015" – please cite corresponding studies.

"Owing to the panicky upsurge in global warming"- same comment.

Currently, the introduction presents a mix of general affirmation and a few results obtained in previous studies.

Please divide lines 45 to 151 into different paragraphs and address corresponding studies in each of them (One paragraph for NaOH concentration effect, one for NaOH/NA2SiO3 effect, one for fly ash, and another for steel slag, then combined in one paragraph that highlight the results presented in the most similar papers). Please conduct a comprehensive and exhaustive study of the previous literature (the latest research, currently the references are old and not enough (there are only one reference to publications from 2020, none from 2021, 2022, 2019 – last five years), please consider the results presented in these studies: DOI: 10.3390/ma15072667; DOI: 10.1016/j.conbuildmat.2022.126843; DOI: 10.3390/ma15010375 ). Please clearly highlight the pros and cons of previous results and justify the need for the current research. Please discuss the highlights individually and assure a clear correspondence between the affirmations from the manuscript and those from the cited papers.

"No researchers have so far reported on this shape of Geopolymer ferrocement panels" There is only one citation from the last five years in the literature (the literature survey wasn’t conducted).

Also, how can be stated that the geopolymeric ferrocement "can withstand heat, sound, fire resistance" if these properties were not tested? Please provide relevant results.

Materials and Methods

Two types of iron oxides have been detected in these types of materials, therefore, please replace Fe2O3 with FexOy or provide scientific proof to support your results. Moreover, which methods/equipment have been used to evaluate the properties presented in the table? Are these data obtained by the authors or they have been provided by the manufacturer (please introduce corresponding comments into the manuscript).

Figure 1. Wire mesh, however, in the content is described as "chicken mech", please use the same terminology. Also, the image is poor, please cut a representative part of the mesh, put it on a white background and provide a picture with a scalebar which is representative of the characteristics of the mesh or cite the production standards/company.

Methods and equipment – each equipment have specific measuring errors. Please provide representative information for each equipment involved in the study.

 Compressive response of Geopolymer Mortar

NaOH molarity optimization- How was this parameter evaluated, what was the value of the other parameters when the effect of NaOH molarity was tested (what was  the ratio between fly ash and steel slag, what was the solid to liquid ratio, was NaOH used as a single activator or it was mixed with Na2SiO3, if yes what was the NaOH/Na2SiO3 ratio?) Please provide corresponding comments on the manuscript. Please clearly describe how these parameters have been optimized.

XRD analysis – There are multiple peaks present in the XRD pattern which haven’t been considered. Why do the authors consider some peaks instead of others (there are some clear peaks around 32, 38, 58, 62, 68 etc.)? Please improve the description of the XRD spectra (see DOI: 10.3390/ma1501020 and DOI: 10.1061/(ASCE)MT.1943-5533.0003012)

Figure 20. Shows the crack pattern of Geopolymer Ferrocement panels. – The images are unclear and the pattern cannot be observed. Please introduce figure labels to highlight the areas of interest for the readers.

Author Response

(The authors gave the same response as above.)

Round 2

Reviewer 2 Report

The authors only state that they considered my suggestions and recommendations. However, the manuscript wasn't revised as suggested. 

The XRD spectra show multiple peaks that weren't identified. Also, the newly described peaks weren't presented in the spectra.

The FexOy was presented in the manuscript as FeO (this type of oxide doesn't exist).

Na2SiO3 was been presented with subscript.

Please revise the manuscript according to my previous suggestions.

Author Response

Please find enclosed the response to the reviewer comments.
